# DISPEL: DOMAIN GENERALIZATION VIA DOMAIN-SPECIFIC LIBERATING

## ABSTRACT

Domain generalization aims to learn a generalization model that can perform well on unseen test domains by only training on limited source domains. However, existing domain generalization approaches often bring in prediction-irrelevant noise or require the collection of domain labels. To address these challenges, we consider the domain generalization problem from a different perspective by categorizing the underlying feature groups into domain-shared and domain-specific features. Nevertheless, domain-specific features are difficult to be identified and distinguished from the input data. In this work, we propose DomaIn-SPEcific Liberating (DISPEL), a post-processing fine-grained masking approach that can filter out undefined and indistinguishable domain-specific features in the embedding space. Specifically, DISPEL utilizes a mask generator that produces a unique mask for each input data to filter domain-specific features. The DISPEL framework is highly flexible to apply to fine-tuned models. We derive a generalization error bound to guarantee the generalization performance by optimizing a designed objective loss. The experimental results on five benchmarks demonstrate that DISPEL outperforms existing methods and can further generalize various algorithms. Our source code is available at: https://anonymous.4open.science/r/dispel-C6FE

## 1 INTRODUCTION

Deep neural network (DNN) models have achieved impressive results in various fields, including object recognition (Wei et al., 2019; Tan et al., 2020a; De Vries et al., 2019; Barbu et al., 2019; Chen et al., 2019), semantic segmentation (Strudel et al., 2021; Cheng et al., 2021; Xie et al., 2021b; Wang et al., 2021; Zheng et al., 2021), and object detection (Dai et al., 2021; Joseph et al., 2021; Tan et al., 2020b; Carion et al., 2020; Chang et al., 2021; Xie et al., 2021a). However, the distribution of test data in real-world applications may be statistically different from training data, causing DNN models to suffer severe performance drops. In this case, domain generalization research aims to address the challenges of distribution difference, known as the domain shifting problem. Domain generalization is essential for

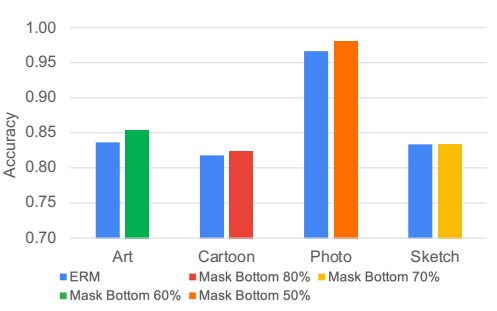

Figure 1: Impact of Global Masking on Unseen PACS Domains (see Sec. 2.2). Mask Bottom p% means the optimal masking levels in the embedding dimensions. We present the most effective p% for each unseen domain.

real-world applications where collecting representative training data may be difficult or costly. Thus, it is crucial to generalize models with limited domain training data.

Existing algorithms for domain generalization can be categorized into two branches: data manipulation and representation learning. The first branch, data manipulation (Tobin et al., 2017; Tremblay et al., 2018; Volpi et al., 2018; Volpi & Murino, 2019; Shi et al., 2020; Xu et al., 2020; Yan et al., 2020; Qiao et al., 2020), focuses on reducing overfitting issues by increasing the diversity and quantity of available training data through data augmentation methods or generative models.

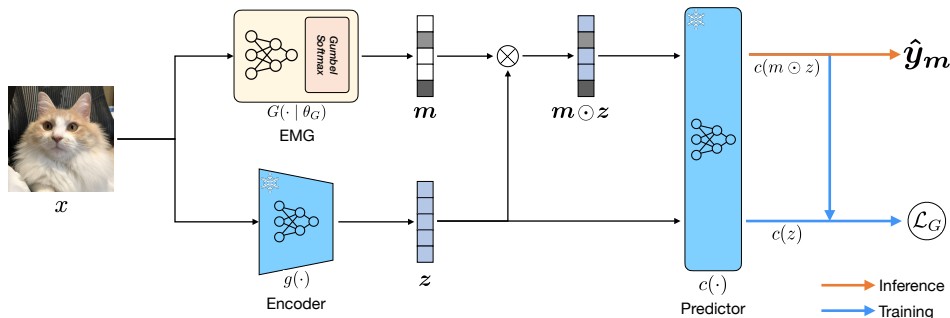

Figure 2: An overview of the proposed framework DISPEL, where EMG refers to Embedding Mask Generator (see Sec. 3.2), and $\mathcal{L}_G$ denotes the objective loss (see Eq. 3). The framework DISPEL first splits a given frozen fine-tuned model into an encoder $g(\cdot)$ and a predictor $c(\cdot)$, and then updates EMG by minimizing cross entropy loss between $c(\boldsymbol{m} \odot \boldsymbol{z})$ and $c(\boldsymbol{z})$.

Two well-known studies in this branch are Mixup (Zhang et al., 2017; Wang et al., 2020) and Randomization (Peng et al., 2018). The second branch, representation learning (Khosla et al., 2012; Huang et al., 2020; Li et al., 2018a;b; Ganin et al., 2016; Li et al., 2018d;c; Chattopadhyay et al., 2020; Piratla et al., 2020; Cha et al., 2022), aims to learn an encoder that generates invariant embeddings among different domains. This branch includes popular algorithms such as CORAL (Sun & Saenko, 2016), IRM (Arjovsky et al., 2019), and DRO (Sagawa et al., 2019). Beyond algorithm design, model selection during training can also influence generalization performance. A recent research (Gulrajani & Lopez-Paz, 2020) proposes two model selection settings in which the selected models can be trained to achieve good generalization results. However, there are several limitations among existing approaches. First, data manipulation methods require highly engineered efforts and can introduce too much prediction-irrelevant knowledge label noise, which will degrade prediction performance. Second, the penalty regularizations of representation learning often require domain labels or information to train the domain-invariant encoder. Existing model selection settings also need domain labels to obtain desirable validation subsets. However, obtaining these domain labels can be a significant expense or may not be feasible in real-world applications.

To address the challenges mentioned above, we consider the domain generalization problem from a different perspective by categorizing the underlying features into two groups, i.e., *domain-shared and domain-specific features*. The domain-shared features generally exist in all domains, and the domain-specific features only exist in certain domains. The prediction models typically experience a drop in performance when tested on unseen domains excluded in training data. This is because the prediction models are trained on both domain-shared and domain-specific features, leading to ineffective prediction outcomes on unseen domain data. Intuitively, we can make a prediction model independent of domain-specific features by eliminating them during the inference stage. However, there are two difficulties in using this solution. First, domain-specific features are usually hard to identify or predefine because those features are represented in different formats. For instance, one domain-specific feature could be the color of objects in the seen domains from the training set. This means that an object's color is highly correlated with its ground truth. Nevertheless, when the domain-specific features are the interweaved relationship between an object's color and stripe, it becomes difficult to recognize the correlation between each domain-specific feature and the ground truth. Second, domain-specific features are typically difficult to be distinguished directly from the original image. Take an "Art style" car image as an example. It is impalpable what is the "Art style" car image, which only includes domain-specific features, and what is the car image without any "style," which only contains domain-shared features.

To address the two difficulties, we perform a post-processing method, a global mask, to filter out domain-specific features in the latent space. We verify its efficacy by designing a naïve global mask to filter out the domain-specific features. The global mask is identified by ranking the Permutation Importance (Fisher et al., 2019) of each embedding dimension. The results in Fig. 1 show that utilizing the global mask can improve the generalization performance of a fine-tuned[1] model on different unseen domains. However, it is a sub-optimal solution since the global mask does not

---

[1]A fine-tuned model refers to the pre-trained model that has been further trained on a specific task using a training set, where the pre-trained model is pre-trained on a large-scale dataset such as ImageNet.

account for the discrepancy among training instances. As illustrated in Fig. 1, the effectiveness of masked dimensions varies for different compositions of training domains.

To account for instance discrepancies among various training domains without relying on domain labels, we propose DomaIn-SPEcific Liberating (DISPEL), a post-processing fine-grained masking approach that extracts out domain-specific features in the embedding space. Our proposed framework is illustrated in Fig. 2. **The key idea of DISPEL is to learn a mask generator that automatically generates a distinct mask for each input data, to filter domain-specific features from the embedding space.** The effectiveness of DISPEL has been demonstrated theoretically and empirically to improve generalization performance without utilizing domain labels. The experimental results on five benchmark datasets indicate that DISPEL achieves state-of-the-art performance, even without leveraging domain labels and any data augmentation method. This performance exceeds that of algorithms that require domain labels for training. Our main contributions are as follows:

- We propose DISPEL to generalize a frozen model by masking out hard-to-identify domain-specific features in the embedding space without adopting domain labels.

- Theoretical analysis guarantees that DISPEL improves the generalization performance of frozen prediction models by minimizing the generalization error bound.

- Experimental results on various benchmarks demonstrate the effectiveness of DISPEL and it can also further improve the existing domain generalization algorithms.

## 2 A Naïve Method with Global Masking

In this section, we will go over the notations used in this paper and demonstrate the efficacy and limitation of a naïve global masking strategy.

### 2.1 The Problem of Domain Generalization

A domain is a collection of data drawn from a distribution. Given a training set $\mathcal{X}$ with a label set $\mathcal{Y}$, the goal of domain generalization is to find a perfect generalized model $f : \mathcal{X} \rightarrow \mathcal{Y}$ with parameter $\theta \in \Theta$ that can perform well on both training domains and unseen test domains. Note that unseen test domains are not included in training domains. Let $\mathcal{D}_i$ be a distribution over input space $\mathcal{X}$ and $\mathcal{X} := \{\mathcal{D}_i\}_{i=1}^I$ be a set of training domains, where $I$ denotes the total number of training domains. We denote each training domain as $\mathcal{D}_i = \{\boldsymbol{x}_j^i\}_{j=1}^{\mathcal{J}}$ and define a set of unseen domain samples $\mathcal{T} := \{\boldsymbol{x}_k^T\}_{k=1}^K$, where $\mathcal{J}$ denotes the number of training domain samples and $K$ represents the number of unseen domain samples.

### 2.2 Globally Masking Domain-Specific Features

The performance of prediction models is typically degraded while testing on unseen domains due to accounting for domain-specific features. We experimentally show that masking out the domain-specific features in the embedding space during inference can improve the generalization ability of prediction models. Specifically, given a prediction model, we split it into an encoder for mapping input data to embedding space, and a predictor for predicting. Then we leverage a feature explainer for calculating importance scores for each embedding dimension, assuming that the less accumulated importance scores among training data indicate the more domain-specific dimensions. Following this idea, a global mask can be found by identifying a certain number of domain-specific dimensions. Then we can leverage it to block out the domain-specific embedding dimensions to make the predictor focus on domain-shared features. In our following experiment, the global masks are obtained by calculating permutation importance (Fisher et al., 2019). The results on the PACS dataset shown in Fig. 1, demonstrating that prediction accuracy on unseen domains of a fine-tuned ERM model can be improved by a global mask blocking out a certain percentage of dimensions.

Despite the efficacy of the global masking method, it is a suboptimal solution because a global mask cannot consider the discrepancy among instances of different domains for each sample. As shown in Fig. 1, the effectiveness of masked embedding dimensions varies among different compositions of training domains. In other words, the improvements raised by a single global mask will vary among different unseen domains.

# 3 DOMAIN-SPECIFIC LIBERATING (DISPEL)

To address the limitation of the global masking method as introduced in Sec. 2.2, we aim to achieve domain generalization by considering a fine-grained masking method. To this end, in this section, we formally propose the DomaIn-SPEcific Liberating (DISPEL) framework that prevents a prediction model from focusing on domain-specific features projected on the dimensions of each embedding.

## 3.1 OVERALL DISPEL FRAMEWORK

The prediction models being fine-tuned with ERM may have limited generalization performance on an unseen test domain. To address this, we introduce DISPEL, a post-processing domain generalization framework that enhances the fine-tuned model's generalization performance without modifying its parameters. Specifically, DISPEL improves its generalization by accounting for instance discrepancies among various training domains without using domain labels. We propose a fine-grained masking component named Embedding Mask Generator (EMG) that can generate instance-specific masks for blocking the domain-specific features in the embedding space (see Fig. 2). The framework of DISPEL is given as follows. Generally, we freeze the prediction model and split it into an encoder $g(\cdot) : \mathcal{X} \to \mathcal{Z}$ and a predictor $c(\cdot) : \mathcal{Z} \to \mathcal{Y}$, where $\mathcal{Z}$ is the embedding space output by $g(\cdot)$. Given an input instance $\boldsymbol{x} \in \mathcal{X}$, EGM can generate a mask $\boldsymbol{m}$ for the embedding $\boldsymbol{z} := g(\boldsymbol{x})$ of the input data to filter domain-specific features. Finally, the frozen model can achieve domain generalization on the unseen test domain via the mask generated by EMG. We will introduce the implementation details in Sec. 4.1. In the following, we will introduce the details and training process of the proposed EMG component.

## 3.2 EMBEDDING MASK GENERATOR (EMG)

Considering the inherent problem among training data from different domains, the Embedding Mask Generator (EMG) $G : \mathcal{X} \to \mathbb{R}^{\mathrm{d}}$ aims to generate an instance-specific mask to mask out domain-specific features corresponding to the input data in embedding space. Specifically, for a data instance $\boldsymbol{x}$, a domain-specific embedding mask is generated to satisfy the $d$-dimensional Binomial distribution $\mathcal{B}(1 - p_1, \cdots, 1 - p_{\mathrm{d}})$, where $[p_1, \cdots, p_{\mathrm{d}}] = G(\boldsymbol{x} \mid \theta_G)$. To enable the updating of the base model of the EMG $G(\cdot \mid \theta_G)$ through backward propagation, a temperature-dependent Gumbel Softmax operation is adopted. Formally, the embedding mask is generated as follows:

$$\boldsymbol{m}_j^i = \frac{\exp\{(\log(\mathbf{1} - \boldsymbol{p}) + \boldsymbol{h})/\tau\}}{\exp\{(\log(\mathbf{1} - \boldsymbol{p}) + \boldsymbol{h})/\tau\} + \exp\{(\log(\boldsymbol{p}) + \boldsymbol{h}')/\tau\}} \quad (1)$$

where $\log(\cdot)$ and $\exp(\cdot)$ denote the element-wise operation of a vector; $\tau$ is a temperature hyperparameter for controlling the degree of discreteness of the generated distribution; and $\boldsymbol{h} = [h_1, \cdots, h_d]$ and $\boldsymbol{h}' = [h'_1, \cdots, h'_d]$ are randomly sampled from $d$-dimensional standard Gumbel distribution $\mathrm{Gumbel}(\mathbf{0}, \mathbf{1})$:

$$\mathrm{Gumbel}(\mathbf{0}, \mathbf{1}) = -\log(-\log u), u \sim \mathrm{Uniform}(0, 1). \quad (2)$$

The $\tau$ in the Gumbel Softmax operation is an implicit regularization that can be used to avoid the trivial outcome of EMG generating masks that filter out no features.

## 3.3 THE TRAINING OF EMG MODULE

The goal of DISPEL is to improve the generalization of a frozen prediction model by preserving domain-shared features while filtering out domain-specific features via $\boldsymbol{m}_j^i \odot \boldsymbol{z}_j^i$. To achieve this goal, the output of the frozen predictor with a masked embedding $c(\boldsymbol{m}_j^i \odot \boldsymbol{z}_j^i) = \hat{\boldsymbol{y}}_m$ should be close to the embedding without being masked $c(\boldsymbol{z}_j^i) = \hat{\boldsymbol{y}}$ for all $\boldsymbol{z}_j^i \in \mathcal{Z}$ and $1 \le j \le \mathcal{J}$. Following this intuition, EMG can be updated by the cross entropy loss $L_{CE}$ between $\hat{\boldsymbol{y}}_m$ and $\hat{\boldsymbol{y}}$ as follows:

$$\mathcal{L}_G = \min_{\theta_G} \sum_{i=1}^{I} \sum_{j=1}^{\mathcal{J}} L_{CE}(\hat{\boldsymbol{y}}_m, \hat{\boldsymbol{y}}). \quad (3)$$

The trained EMG $G(\cdot \mid \theta_G)$ is a domain-specific feature mask generator across multiple domains, which does not require domain labels for training.

### 3.4 GENERALIZATION BOUND OF DISPEL

The objective function of EMG module in Eq. 3 aims to mitigate the performance drop on unseen test domains, which is caused by domain-specific features. The generalization error is thereby amply due to the impact of domain-specific features. In this section, we provide Theorem 1 to show that the generalization error can be effectively bounded by DISPEL, referring to a better DG capability across the unseen domains.

**Theorem 1 (Generalization Error Bound).** *Let $\widetilde{x}_k^T$ be a masked instance of $x_k^T$ on an unseen domain $T$. Given an instance embedding $z_k^T$ satisfies the composition of domain-specific $z_k^{T\text{-}sh}$ and domain-sharing $z_k^{T\text{-}sp}$, where $\hat{f}(x_k^T) = \hat{c}(z_k^T)$ be the predicted outcomes. The generalization error $GE = \mathbf{E}_{\mathcal{X}}[\,\|f(x_k^T), \hat{f}(\widetilde{x}_k^T)\|_2\,]$ of DISPEL framework can be bounded as:*

$$GE \leq \mathbf{E}_{\mathcal{X}}[\|c(z_k^{T\text{-}sh}) - \hat{c}(\widetilde{z}_k^{T\text{-}sh})\|_2] + \mathbf{E}_{\mathcal{X}}[\|c(z_k^{T\text{-}sp}) - \hat{c}(\widetilde{z}_k^{T\text{-}sp})\|_2] \tag{4}$$

*where $\widetilde{z}_k^T = z_k^T \odot m_k^T$ is composed of remained domain-specific embedding $\widetilde{z}_k^{T\text{-}sp}$ and preserved domain-sharing embedding $\widetilde{z}_k^{T\text{-}sh}$.*

Theorem 1 shows that the upper bound on the GE depends on two terms, the predicted error of *domain-shared features* and of *domain-specific features*. The proof of Theorem 1 is provided in Appendix A. Based on Eq. 1, DISPEL contributes to minimizing the upper bound of GE as follows:

- The objective in Eq. 3 aligns with the goal of minimizing the first term of Theorem 1, as $m_k^T$ aims to preserve the domain-shared features on $z_k^{\text{T-sh}}$ for the prediction of multiple training domains.

- DISPEL framework minimizes the value of second terms in Eq. 1 by making $\widetilde{z}_k^{\text{T-sp}}$ approaches $z_k^{\text{T-sp}}$ with a generated mask.

### 3.5 ALGORITHM OF EMG TRAINING

The training outline of the EMG is given in Algorithm 1. The training aims to achieve the base model of EMG $G(\cdot \mid \theta_G)$ that can generate domain-specific feature masks for each input data. Specifically, EMG generates instance-specific embedding masks based on given training data (line 4), and then we let the frozen predictor $c(\cdot)$ predicts given masked embedding and original embedding (line 5-6). In each iteration, EMG is updated according to Eq. 3 (line 7) until it converges.

---

**Algorithm 1** Embedding Mask Generator (EMG) Training

---

1: **Input:**
      Training dataset $x \in \mathcal{X}$
      Frozen feature encoder $g(x) = z$
      Frozen predictor $c(\cdot)$
      Base model of EMG $G(\cdot \mid \theta_G)$
2: **Output:**
      Instance-specific mask generator EMG $G(\cdot \mid \theta_G)$
3: **while** not convergence **do**
4:     Generate the embedding mask $m$ by $G(x \mid \theta_G)$ and Eq. 1
5:     Predict by $c(\cdot)$ given masked embedding $\hat{y}_m = c(m \odot z)$
6:     Predict by $c(\cdot)$ given original embedding $\hat{y} = c(z)$
7:     Update $G(\cdot \mid \theta_G)$ by minimizing the objective loss Eq. 3
8: **end while**

---

## 4 EXPERIMENTS

In this section, we conduct experiments to evaluate the performance of DISPEL, aiming to answer the following three research questions: **RQ1:** How effective is DISPEL when compared to state-of-the-art baselines? **RQ2:** How does the fine-grained masking manner influence generalization performance? **RQ3:** Can DISPEL improve the generalization of other algorithms?

Table 1: Average unseen domain results (ResNet50).

| | PACS | Office-Home | VLCS | TerraInc | DomainNet | Avg. |
|---|---|---|---|---|---|---|
| **Group 1**: algorithms requiring domain labels | | | | | | |
| Mixup (Wang et al., 2020) | $87.7 \pm 0.5$ | $71.2 \pm 0.1$ | $77.7 \pm 0.4$ | $48.9 \pm 0.8$ | $39.6 \pm 0.1$ | 65.1 |
| MLDG (Li et al., 2018a) | $84.8 \pm 0.6$ | $68.2 \pm 0.1$ | $77.1 \pm 0.4$ | $46.1 \pm 0.8$ | $41.8 \pm 0.4$ | 63.6 |
| CORAL (Sun & Saenko, 2016) | $86.2 \pm 0.2$ | $70.1 \pm 0.4$ | $77.7 \pm 0.5$ | $46.4 \pm 0.8$ | $41.8 \pm 0.2$ | 64.4 |
| MMD (Li et al., 2018b) | $87.1 \pm 0.2$ | $70.4 \pm 0.1$ | $76.7 \pm 0.9$ | $49.3 \pm 1.4$ | $39.4 \pm 0.8$ | 64.6 |
| DANN (Ganin et al., 2016) | $86.7 \pm 1.1$ | $69.5 \pm 0.6$ | $78.7 \pm 0.3$ | $48.4 \pm 0.5$ | $38.4 \pm 0.0$ | 64.3 |
| C-DANN (Li et al., 2018d) | $82.8 \pm 1.5$ | $65.6 \pm 0.5$ | $78.2 \pm 0.4$ | $47.6 \pm 0.8$ | $38.9 \pm 0.1$ | 62.6 |
| DA-ERM (Dubey et al., 2021) | $84.1 \pm 0.5$ | $67.9 \pm 0.4$ | $78.0 \pm 0.2$ | $47.3 \pm 0.5$ | $43.6 \pm 0.3$ | 64.2 |
| **Group 2**: algorithms without requiring domain labels | | | | | | |
| ERM (Vapnik, 1999) | $86.4 \pm 0.1$ | $69.9 \pm 0.1$ | $77.4 \pm 0.3$ | $47.2 \pm 0.4$ | $41.2 \pm 0.2$ | 64.5 |
| IRM (Arjovsky et al., 2019) | $84.4 \pm 1.1$ | $66.6 \pm 1.0$ | $78.1 \pm 0.0$ | $47.9 \pm 0.7$ | $35.7 \pm 1.9$ | 62.5 |
| DRO (Sagawa et al., 2019) | $86.8 \pm 0.4$ | $70.2 \pm 0.3$ | $77.2 \pm 0.6$ | $47.0 \pm 0.3$ | $33.7 \pm 0.2$ | 63.0 |
| RSC (Huang et al., 2020) | $86.9 \pm 0.2$ | $69.4 \pm 0.4$ | $75.3 \pm 0.5$ | $45.7 \pm 0.3$ | $41.2 \pm 1.0$ | 63.7 |
| MIRO (Cha et al., 2022) | $85.4 \pm 0.4$ | $70.5 \pm 0.4$ | $79.0 \pm 0.0$ | $50.4 \pm 1.1$ | $\mathbf{44.3} \pm 0.2$ | 65.9 |
| DISPEL | $\mathbf{88.2} \pm 0.1$ | $\mathbf{73.3} \pm 0.3$ | $\mathbf{79.3} \pm 0.1$ | $\mathbf{50.4} \pm 0.2$ | $44.1 \pm 0.0$ | $\mathbf{67.1}$ |

## 4.1 EXPERIMENTAL SETTINGS

**Datasets.** To compare the efficacy of our proposed framework with existing algorithms, we conduct our experiments on five real-world benchmark datasets: PACS (Li et al., 2017) with 7 classes of images in 4 domains, Office-Home (Venkateswara et al., 2017) with 65 classes of images in 4 domains, VLCS (Fang et al., 2013) with 5 classes of images in 4 domains, Terra Incognita (Beery et al., 2018) with 10 classes of images in 4 domains, and DomainNet (Peng et al., 2019) with 345 classes of images in 6 domains. DomainNet is a larger-scale dataset with a more difficult multi-classification task than the other 4 benchmark datasets. Details of the datasets can be found in Appendix C.

**Baselines.** To fairly compare our proposed framework with existing algorithms, we follow the settings of DomainBed (Gulrajani & Lopez-Paz, 2020) and DeepDG (Wang et al., 2022), using the best result between DomainBed, DeepDG, and the original literature. We categorize the 12 baseline algorithms into two groups: **Group 1**: the algorithms requiring domain labels (Mixup (Wang et al., 2020), MLDG (Li et al., 2018a), CORAL (Sun & Saenko, 2016), MMD (Li et al., 2018b), DANN (Li et al., 2018d), C-DANN (Li et al., 2018d), and DA-ERM (Dubey et al., 2021)); and **Group 2**: the algorithms without requiring domain labels (ERM (Vapnik, 1999), IRM (Arjovsky et al., 2019), DRO (Sagawa et al., 2019), RSC (Huang et al., 2020), and MIRO (Cha et al., 2022)). More baseline details can be found in Appendix D.

**Implementation Details.** Our experiments leverage the DeepDG codebase (Wang et al., 2022) and employ a training-domain validation set for model selection. We primarily use ResNet50 for all five benchmarks, with ResNet18 results provided for reference in Appendix E. Both ResNet architectures are fine-tuned by ERM, as detailed in Sec. 3.1. For the EMG module in DISPEL, we use a pre-trained ResNet50 base model and set the temperature hyper-parameter $\tau$ in Eq. 1 to 0.1. Additional implementation details are in Appendix D.

## 4.2 GENERALIZATION EFFICACY ANALYSIS (RQ1)

We evaluate the generalization ability of DISPEL against 12 baselines across 5 datasets, summarizing the ResNet50-based results in Tab. 1. Overall, DISPEL achieves state-of-the-art without using domain labels. DISPEL achieves the best average accuracy on unseen test domains in 4 out of 5 datasets. On average, DISPEL shows the best accuracy on unseen test domains over 5 benchmarks, indicating stable effectiveness across different data distributions. Considering the extensive experimentation conducted on 5 benchmark datasets and 22 unseen test domains, the results conclusively demonstrate the efficacy of DISPEL in improving the diverse types of image data. The full experimental results of DISPEL can be found in Appendix E.

**Observation 1: DISPEL can achieve state-of-the-art based on a frozen fine-tuned ERM model.** Based on the results shown in Tab. 1, DISPEL outperforms other existing methods by generalizing a frozen ERM prediction model without requiring domain labels.

Table 2: Each unseen test domain accuracy comparisons of PACS (ResNet50).

| | Art Painting | Cartoon | Photo | Sketch |
|---|---|---|---|---|
| **Group 1**: algorithms requiring domain labels | | | | |
| Mixup (Wang et al., 2020) | **89.3** ± 0.5 | 81.7 ± 0.1 | 97.3 ± 0.4 | 82.3 ± 0.8 |
| MLDG (Li et al., 2018a) | 89.1 ± 0.9 | 78.8 ± 0.7 | 97.0 ± 0.9 | 74.4 ± 2.0 |
| CORAL (Sun & Saenko, 2016) | 84.7 ± 0.6 | 81.5 ± 1.1 | 96.7 ± 0.0 | 81.7 ± 0.1 |
| MMD (Li et al., 2018b) | 84.5 ± 0.6 | 79.7 ± 0.7 | 97.5 ± 0.4 | 78.1 ± 1.3 |
| DANN (Ganin et al., 2016) | 87.1 ± 0.5 | **83.2** ± 1.4 | 96.5 ± 0.2 | 79.8 ± 2.8 |
| C-DANN (Li et al., 2018d) | 84.0 ± 0.9 | 78.5 ± 1.5 | 97.0 ± 0.4 | 71.8 ± 3.9 |
| **Group 2**: algorithms without requiring domain labels | | | | |
| ERM (Vapnik, 1999) | 83.7 ± 0.1 | 81.8 ± 1.3 | 96.7 ± 0.0 | 83.4 ± 0.9 |
| IRM (Arjovsky et al., 2019) | 85.0 ± 1.6 | 77.6 ± 0.9 | 96.7 ± 0.3 | 78.5 ± 2.6 |
| DRO (Sagawa et al., 2019) | 85.0 ± 0.3 | 81.8 ± 0.8 | 96.1 ± 0.3 | 84.3 ± 0.7 |
| RSC (Huang et al., 2020) | 87.8 ± 0.8 | 80.3 ± 1.8 | 97.7 ± 0.3 | 81.5 ± 1.2 |
| DISPEL | 87.1 ± 0.1 | 82.5 ± 0.0 | **98.0** ± 0.1 | **85.2** ± 0.1 |

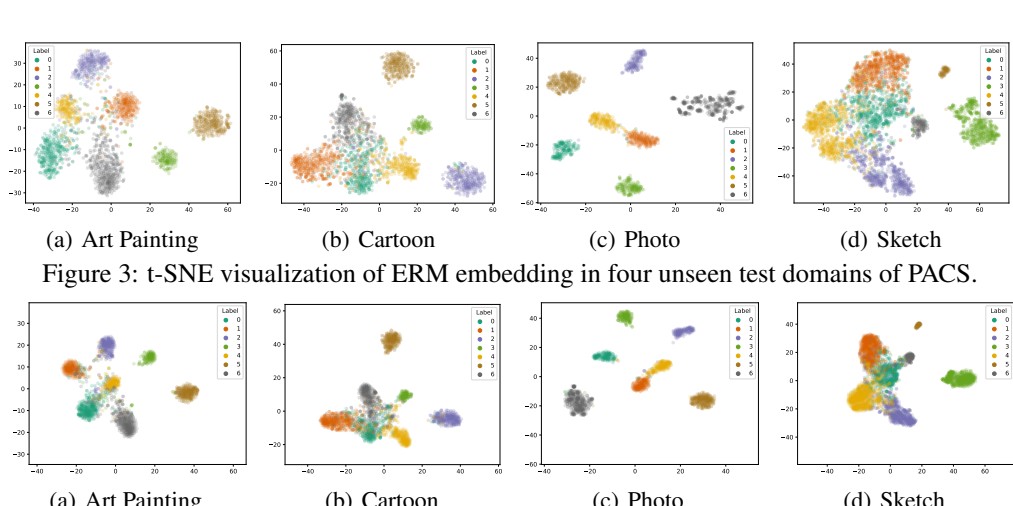

| (a) Art Painting | (b) Cartoon | (c) Photo | (d) Sketch |
|---|---|---|---|

Figure 3: t-SNE visualization of ERM embedding in four unseen test domains of PACS.

| (a) Art Painting | (b) Cartoon | (c) Photo | (d) Sketch |
|---|---|---|---|

Figure 4: t-SNE visualization of DISPEL embedding in four unseen test domains of PACS.

## 4.3 MANNERS OF DOMAIN-SPECIFIC LIBERATING (RQ2)

DISPEL aims to enhance the generalizability of fine-tuned models in a fine-grained instance-specific manner, which considers the unique characteristic of each instance in masking embedding. In addition to the encouraging experimental outcomes of DISPEL, we delve into the detailed mechanisms underlying DISPEL's performance.

### 4.3.1 OOD PERFORMANCE IN EACH UNSEEN TEST DOMAIN

We first observe the prediction accuracy in each unseen test domain. DISPEL can improve the accuracy of the base algorithm (e.g., ERM) on 21 out of 22 unseen test domains, except the domain Location 43 of the Terra Incognita dataset. Tab. 2 shows the accuracy of each unseen test domain of PACS. Despite only achieving the best performance in two unseen test domains, DISPEL can increase the accuracy in all four domains compared to the base algorithm ERM. This phenomenon reflects the instance-specific manner of DISPEL due to the ability to mask data representations according to each input image, which is a more fine-grained perspective than identifying different domain groups for generalization.

Regarding the Terra Incognita, the results show that DISPEL significantly enhances the accuracy of ERM except for *Location 43* domain of Terra Incognita. DISPEL can still averagely outperform four unseen test domains of Terra Incognita. The results again show that the fine-grained instance-specific embedding masking of DISPEL brings a more stable generalizing efficacy among different out-of-distribution sets. Considering the DoainNet as a larger-scale benchmark with a more difficult multi-classification task, DISPEL achieves the best accuracy in five out of six unseen test domains and improves the accuracy in all the unseen domains compared to ERM, as shown in Tab. 3.

Table 3: Each unseen test domain accuracy comparisons of DomainNet (ResNet50).

| | Clipart | Infograph | Painting | Quickdraw | Real | Sketch |
|---|---|---|---|---|---|---|
| **Group 1**: algorithms requiring domain labels | | | | | | |
| Mixup (Wang et al., 2020) | $55.3 \pm 0.3$ | $18.2 \pm 0.3$ | $45.0 \pm 1.0$ | $12.5 \pm 0.3$ | $57.1 \pm 1.2$ | $49.2 \pm 0.3$ |
| MLDG (Li et al., 2018a) | $59.5 \pm 0.0$ | $19.8 \pm 0.4$ | $\mathbf{48.3} \pm 0.5$ | $13.0 \pm 0.4$ | $59.5 \pm 1.0$ | $50.4 \pm 0.7$ |
| CORAL (Sun & Saenko, 2016) | $58.7 \pm 0.2$ | $\mathbf{20.9} \pm 0.3$ | $47.3 \pm 0.3$ | $13.6 \pm 0.3$ | $60.2 \pm 0.3$ | $50.2 \pm 0.6$ |
| MMD (Li et al., 2018b) | $54.6 \pm 1.7$ | $19.3 \pm 0.3$ | $44.9 \pm 1.1$ | $11.4 \pm 0.5$ | $59.5 \pm 0.2$ | $47.0 \pm 1.6$ |
| DANN (Ganin et al., 2016) | $53.8 \pm 0.7$ | $17.8 \pm 0.3$ | $43.5 \pm 0.3$ | $11.9 \pm 0.5$ | $56.4 \pm 0.3$ | $46.7 \pm 0.5$ |
| C-DANN (Li et al., 2018d) | $53.4 \pm 0.4$ | $18.3 \pm 0.7$ | $44.8 \pm 0.3$ | $12.9 \pm 0.2$ | $57.5 \pm 0.4$ | $46.7 \pm 0.2$ |
| **Group 2**: algorithms without requiring domain labels | | | | | | |
| ERM (Vapnik, 1999) | $58.4 \pm 0.3$ | $19.2 \pm 0.4$ | $46.3 \pm 0.5$ | $12.8 \pm 0.0$ | $60.6 \pm 0.5$ | $49.7 \pm 0.8$ |
| IRM (Arjovsky et al., 2019) | $51.0 \pm 3.3$ | $16.8 \pm 1.0$ | $38.8 \pm 2.1$ | $11.8 \pm 0.5$ | $51.5 \pm 3.6$ | $44.2 \pm 3.1$ |
| DRO (Sagawa et al., 2019) | $47.8 \pm 0.6$ | $17.1 \pm 0.6$ | $36.6 \pm 0.7$ | $8.8 \pm 0.4$ | $51.5 \pm 0.6$ | $40.7 \pm 0.3$ |
| RSC (Huang et al., 2020) | $55.0 \pm 1.2$ | $18.3 \pm 0.5$ | $44.4 \pm 0.6$ | $12.2 \pm 0.6$ | $55.7 \pm 0.7$ | $47.8 \pm 0.9$ |
| DISPEL | $\mathbf{63.4} \pm 0.1$ | $20.1 \pm 0.0$ | $\mathbf{48.2} \pm 0.0$ | $\mathbf{14.2} \pm 0.0$ | $\mathbf{63.4} \pm 0.0$ | $\mathbf{54.9} \pm 0.0$ |

**Observation 2: DISPEL possesses stable generalizing efficacy.** The results show that DISPEL maintains its stable efficacy in improving generalization ability over more different data distributions in more diverse classes of data. These results also reflect the purpose of the EMG module that considers each instance for fine-grained domain-specific feature masking.

### 4.3.2 VISUALIZATION ANALYSIS VIA t-SNE

To illustrate how DISPEL improves generalization by blocking domain-specific features in the embedding space, we use t-SNE shown in Fig. 3 and 4 in the unseen test domains of PACS and Terra Incognita by comparing the embedding distribution with and without DISPEL.

Comparing Fig. 3-(a) and Fig. 4-(a), the key observation is that DISPEL aims to make each class more concentrated and separate them better. By drawing down more precise decision boundaries, the predictor can achieve better accuracy in the unseen *Art Painting* domain, where DISPEL enhances the most accuracy among the 4 domains as shown in Tab. 2. Even when the boundaries between classes are blurred and hard to be separated, DISPEL keeps its impact by making them more concentrated inside each class. For instance, the results on *Sketch* domain, as shown in Fig. 3-(d) and Fig. 4-(d), show that DISPEL concentrates the representation of each class, decreasing the length of the boundaries between class 0 and 1, 2, and 4. By decreasing the length of boundaries between different groups, the vague part for classification will be less than the original distribution, which means that the correctness of the predictions in the unseen test domains will increase.

**Observation 3: DISPEL concentrates the distribution of each class embedding.** The t-SNE analysis shows the superiority of DISPEL, improving the domain generalization ability of the fine-tuned ERM by concentrating the distribution of embeddings in the same class. As shown in Tab. 1 to Tab. 3 and Appendix E, DISPEL can then improve classification accuracy in unseen domains.

### 4.3.3 VISUALIZATION OF DISPEL MASKS

To investigate the differences between the global mask (see Sec. 2.2) and the instance-specific masks generated by DISPEL, we provide the visualization in Fig. 5 for the comparison among different classes in *sketch* domain of PACS dataset.

**Observation 4: DISPEL achieve generalization in a fine-grained manner.** As depicted in Fig. 5, DISPEL generates instance-specific masks tailored to each instance, offering a more refined strategy compared to an identical global mask.

### 4.4 BOOSTING OTHER ALGORITHMS VIA DISPEL (RQ3)

In previous experiments, we adopt fine-tuned ERM as the base model of EMG module in DISPEL. Based on the experimental results showing that the proposed DISPEL framework can improve the prediction performance in unseen domains, we are curious if DISPEL can also enhance the generalization ability of other algorithms.

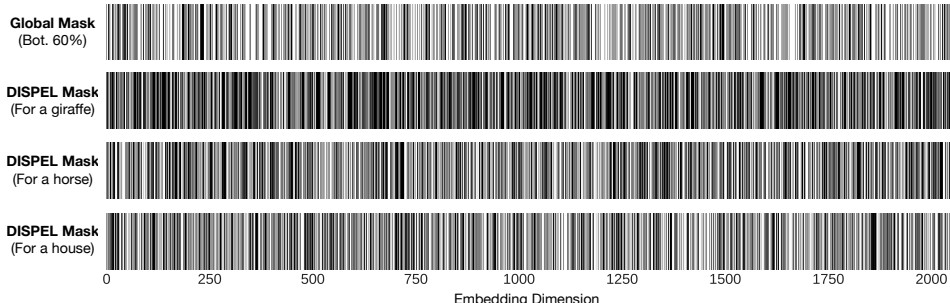

Figure 5: The global mask and DISPEL masks visualization in *sketch* domain of PACS.

Table 4: Average unseen test domain accuracy of DISPEL by using other domain generalization algorithms as baselines. It indicates that DISPEL can boost generalization of other algorithms.

|  | PACS | Office-Home | Avg. |
|---|---|---|---|
| ERM (Vapnik, 1999) | $86.4 \pm 0.1$ | $69.9 \pm 0.1$ | 78.2 |
| ERM w/ DISPEL | $\mathbf{88.2} \pm 0.1$ | $\mathbf{73.3} \pm 0.3$ | **80.8** |
| DRO (Sagawa et al., 2019) | $86.8 \pm 0.4$ | $70.2 \pm 0.3$ | 78.5 |
| DRO w/ DISPEL | $\mathbf{88.2} \pm 0.1$ | $\mathbf{72.2} \pm 0.1$ | **80.2** |
| CORAL (Sun & Saenko, 2016) | $86.2 \pm 0.2$ | $70.1 \pm 0.4$ | 78.2 |
| CORAL w/ DISPEL | $\mathbf{87.5} \pm 0.1$ | $\mathbf{72.8} \pm 0.2$ | **80.2** |
| DANN (Ganin et al., 2016) | $86.7 \pm 1.1$ | $69.5 \pm 0.6$ | 78.1 |
| DANN w/ DISPEL | $\mathbf{87.2} \pm 0.0$ | $\mathbf{72.9} \pm 0.3$ | **80.1** |
| Mixup (Wang et al., 2020) | $87.7 \pm 0.5$ | $71.2 \pm 0.1$ | 79.5 |
| Mixup w/ DISPEL | $\mathbf{89.1} \pm 0.1$ | $\mathbf{73.4} \pm 0.1$ | **81.3** |

We evaluate DISPEL incorporating various baselines, including three representation learning methods—DRO (without requiring domain labels), and CORAL and DANN (requiring domain labels)—as well as a data manipulation method, Mixup (requiring domain labels). As shown in Tab. 4, DISPEL consistently improves prediction accuracy compared to baseline models across two benchmarks. The improvement of DANN w/ DISPEL is substantial in Office-Home but marginal in PACS, as varying initial encoder states may affect different model weights during training.

Additionally, we observe that the original accuracy of an algorithm doesn't indicate its post-DISPEL boost performance. In PACS, Mixup leads and further benefits from DISPEL boosting. However, in Office-Home, DISPEL brings DANN close to ERM and Mixup levels, while its original performance of DRO is slightly behind.

**Observation 5: DISPEL can improve generalization ability for different types of algorithms.** The experimental results of incorporating DISPEL in four algorithms demonstrate that DISPEL can improve the generalization performance of fine-tuned models. According to the results shown in Tab. 4, despite Mixup achieving the best performance with DISPEL, ERM can achieve equivalent results via DISPEL without using domain labels during training.

## 5 CONCLUSIONS AND FUTURE WORK

In this work, we demonstrate the efficacy of masking domain-specific features in embedding space for improving the generalization ability of a fine-tuned prediction model. Based on this observation, we propose a post-processing fine-grained masking framework, DISPEL, by accounting for instance discrepancies to improve generalization ability. Specifically, DISPEL uses a mask generator that generates a distinct mask for each input data, which is used to filter domain-specific features in embedding space. The results on five benchmarks demonstrate that DISPEL outperforms the state-of-the-art baselines. Regarding future directions, we plan to explore the potential of exploiting DISPEL for different downstream tasks and various types of input data. We will consider the characteristics of each task and input data, allowing us to design an objective loss that is tailored to their specific requirements.

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

APPENDIX

## A    PROOF OF THEOREM 1

**Theorem 1** (**Generalization Error Bound**). *Let $\widetilde{\boldsymbol{x}}_k^T$ be a masked instance of $\boldsymbol{x}_k^T$ on an unseen domain $T$. Given an instance embedding $\boldsymbol{z}_k^T$ satisfies the composition of domain-specific $\boldsymbol{z}_k^{T\text{-}sh}$ and domain-sharing $\boldsymbol{z}_k^{T\text{-}sp}$, where $\hat{f}(\boldsymbol{x}_k^T) = \hat{c}(\boldsymbol{z}_k^T)$ be the predicted outcomes. The generalization error $\boldsymbol{GE} = \mathbf{E}_{\mathcal{X}}[\,\|f(\boldsymbol{x}_k^T), \hat{f}(\widetilde{\boldsymbol{x}}_k^T)\|_2\,]$ of DISPEL framework can be bounded as:*

$$\boldsymbol{GE} \leq \mathbf{E}_{\mathcal{X}}[\|c(\boldsymbol{z}_k^{T\text{-}sh}) - \hat{c}(\widetilde{\boldsymbol{z}}_k^{T\text{-}sh})\|_2] + \mathbf{E}_{\mathcal{X}}[\|c(\boldsymbol{z}_k^{T\text{-}sp}) - \hat{c}(\widetilde{\boldsymbol{z}}_k^{T\text{-}sp})\|_2] \tag{5}$$

*where $\widetilde{\boldsymbol{z}}_k^T = \boldsymbol{z}_k^T \odot m_k^T$ is composed of remained domain-specific embedding $\widetilde{\boldsymbol{z}}_k^{T\text{-}sp}$ and preserved domain-sharing embedding $\widetilde{\boldsymbol{z}}_k^{T\text{-}sh}$.*

*Proof.* In order to estimate the generalization error of DISPEL on unseen domain $\mathcal{T}$, we calculate the expected values of distance between $f(\boldsymbol{x}_k^T)$ and $\hat{f}(\widetilde{\boldsymbol{x}}_k^T)$. Without loss of generality, we consider $\ell_2$ norm to evaluate the distance. Hence, the estimated generalization error of $\mathcal{T}$ can be elaborated as:

$$\begin{aligned}
\mathbf{GE} &= \mathbf{E}_{\mathcal{T}}[\,\|f(\boldsymbol{x}_k^T) - \hat{f}(\widetilde{\boldsymbol{x}}_k^T)\|_2\,] \\
&= \int_{\mathcal{X}} \|f(\boldsymbol{x}_k^T) - \hat{f}(\widetilde{\boldsymbol{x}}_k^T)\|_2 P(\mathcal{T})\, d\mathcal{T}
\end{aligned} \tag{6}$$

where $P(\mathcal{T})$ denotes probability density function of $\mathcal{T}$. As a lower generalized error **GE** represents better generalization capability, we can observe from Eq. 6 that the closer $\|f(\boldsymbol{x}_k^T) - \hat{f}(\widetilde{\boldsymbol{x}}_k^T)\|_2$ approaches zero, the better generalization capability is obtained.

In this manner, we now discuss the upper bound of $\|f(\boldsymbol{x}_k^T) - \hat{f}(\widetilde{\boldsymbol{x}}_k^T)\|_2$. This also ensure the upper bound of **GE**. Following the properties that each instance's embedding $\boldsymbol{z}_k^T$ can be composed of domain-specific $\boldsymbol{z}_k^{T\text{-sh}}$ and domain-sharing $\boldsymbol{z}_k^{T\text{-sp}}$, we consider upper bound as follows,

$$\begin{aligned}
&\|f(\boldsymbol{x}_k^T) - \hat{f}(\widetilde{\boldsymbol{x}}_k^T)\|_2 \\
&= \|c(\boldsymbol{z}_k^T) - \hat{f}(\widetilde{\boldsymbol{z}}_k^T)\|_2 \\
&= \|c(\boldsymbol{z}_k^{T\text{-sh}} + \boldsymbol{z}_k^{T\text{-sp}}) - \hat{c}(\widetilde{\boldsymbol{z}}_k^{T\text{-sh}} + \widetilde{\boldsymbol{z}}_k^{T\text{-sp}})\|_2
\end{aligned} \tag{7}$$

Since $c(\cdot)$ is the linear predictor, we can now recast the Eq. 7 in the following,

$$\begin{aligned}
&\|f(\boldsymbol{x}_k^T) - \hat{f}(\widetilde{\boldsymbol{x}}_k^T)\|_2 \\
&= \|c(\boldsymbol{z}_k^{T\text{-sh}} + \boldsymbol{z}_k^{T\text{-sp}}) - \hat{c}(\widetilde{\boldsymbol{z}}_k^{T\text{-sh}} + \widetilde{\boldsymbol{z}}_k^{T\text{-sp}})\|_2 \\
&= \|c(\boldsymbol{z}_k^{T\text{-sh}}) + c(\boldsymbol{z}_k^{T\text{-sp}}) - \hat{c}(\widetilde{\boldsymbol{z}}_k^{T\text{-sh}}) - \hat{c}(\widetilde{\boldsymbol{z}}_k^{T\text{-sp}})\|_2 \\
&\leq \|c(\boldsymbol{z}_k^{T\text{-sh}}) - \hat{c}(\widetilde{\boldsymbol{z}}_k^{T\text{-sh}})\|_2 + \|c(\boldsymbol{z}_k^{T\text{-sp}}) - \hat{c}(\widetilde{\boldsymbol{z}}_k^{T\text{-sp}})\|_2
\end{aligned} \tag{8}$$

Following the conclusion of Eq. 6 and Eq. 8, we have the upper bound of **GE** as follows:

$$\begin{aligned}
\mathbf{GE} &= \mathbf{E}_{\mathcal{T}}[\,\|f(\boldsymbol{x}_k^T) - \hat{f}(\widetilde{\boldsymbol{x}}_k^T)\|_2\,] \\
&= \int_{\mathcal{X}} \|f(\boldsymbol{x}_k^T) - \hat{f}(\widetilde{\boldsymbol{x}}_k^T)\|_2 P(\mathcal{T})\, d\mathcal{T} \\
&\leq \int_{\mathcal{X}} \Big[ \|c(\boldsymbol{z}_k^{T\text{-sh}}) - \hat{c}(\widetilde{\boldsymbol{z}}_k^{T\text{-sh}})\|_2 + \|c(\boldsymbol{z}_k^{T\text{-sp}}) - \hat{c}(\widetilde{\boldsymbol{z}}_k^{T\text{-sp}})\|_2 \Big] P(\mathcal{T}) d\mathcal{T} \\
&= \int_{\mathcal{X}} \|c(\boldsymbol{z}_k^{T\text{-sh}}) - \hat{c}(\widetilde{\boldsymbol{z}}_k^{T\text{-sh}})\|_2\, P(\mathcal{T})\, d\mathcal{T} + \\
&\qquad \int_{\mathcal{X}} \|c(\boldsymbol{z}_k^{T\text{-sp}}) - \hat{c}(\widetilde{\boldsymbol{z}}_k^{T\text{-sp}})\|_2\, P(\mathcal{T})\, d\mathcal{T} \\
&= \mathbf{E}_{\mathcal{X}}[\|c(\boldsymbol{z}_k^{T\text{-sh}}) - \hat{c}(\widetilde{\boldsymbol{z}}_k^{T\text{-sh}})\|_2] + \mathbf{E}_{\mathcal{X}}[\|c(\boldsymbol{z}_k^{T\text{-sp}}) - \hat{c}(\widetilde{\boldsymbol{z}}_k^{T\text{-sp}})\|_2]
\end{aligned}$$

$\square$

## B  RELATED WORKS

There are two primary branches of research in the field of domain generalization: data manipulation and representation learning.

**Data Manipulation.**  The data manipulation branch aims to reduce overfitting by increasing the diversity and quantity of available training data. This is typically achieved through the use of data augmentation methods or generative models (Tobin et al., 2017; Peng et al., 2018; Tremblay et al., 2018; Volpi et al., 2018; Zhang et al., 2017; Xu et al., 2020; Yan et al., 2020; Wang et al., 2020; Yu et al., 2023).

**Representation Learning.**  Representation learning is another branch of methods that focuses on training an encoder that maps samples to a latent space where the embedding remains invariant to various domains (Arjovsky et al., 2019; Sagawa et al., 2019; Huang et al., 2020; Li et al., 2018a;b; Ganin et al., 2016; Cha et al., 2022). Alternative approaches for achieving invariant learning have been proposed, including techniques such as correlation alignment (Sun & Saenko, 2016), class-conditional adversarial learning (Li et al., 2018c), minimizing maximum mean discrepancy (Li et al., 2018d), and mutual information regularization (Cha et al., 2022) that doesn't require domain labels.

**Ensemble Learning.**  There are some ensemble approaches for domain generalization, which train multiple models and then combine the predictions of these models at validation time to obtain a most generalization model. For instance, SWAD (Cha et al., 2021) aims to find a flatter minima and suffers less from overfitting than vanilla SWA (Izmailov et al., 2018) by a dense and overfit-aware stochastic weight sampling strategy; EoA (Arpit et al., 2022) finds that an ensemble of moving average models outperforms a traditional ensemble of unaveraged models.

## C  DATASETS DETAILS

To compare the efficacy of our proposed framework with existing algorithms, we conduct our experiments on 5 real-world benchmark datasets: PACS (Li et al., 2017), Office-Home (Venkateswara et al., 2017), VLCS (Fang et al., 2013), Terra Incognita (Beery et al., 2018), and DomainNet (Peng et al., 2019). Specifically, PACS includes four image styles (Photo, Art, Cartoon, and Sketch), which are considered 4 different domains, and each domain has 7 classes of images (Dog, Elephant, Giraffe, Horse, Person, Guitar, and House) for training and testing. It contains a total of $9,991$ instances in 4 domains. Office-Home consists of 65 classes of images for training and testing. These images belong to four image styles (Art, Clipart, Product, Real) being considered as 4 different domains. It contains a total of $15,588$ instances in 4 domains. VLCS includes images collected from 4 different datasets (Caltech101, LabelMe, SUN09, and VOC2007), which are considered 4 different domains, and each domain has 5 classes (Dog, Bird, Person, Car, and Chair) for training and testing. It contains a total of $10,729$ instances in 4 domains. Terra Incognita consists of 10 classes of photographs of wild animals taken at 4 different locations (Location 100, Location 38, Location 43, and Location 46), considered as 4 different domains. For our experiments, we use the downloader of DomainBed (Gulrajani & Lopez-Paz, 2020) to download the same version Terra Incognita dataset as theirs. It contains a total of $24,788$ instances in 4 domains. DomainNet includes 6 image styles (Clipart, Infograph, Painting, Quickdraw, Real, Sketch) considered as 6 different domains. In each domain, there are 345 classes for training and testing. It contains a total of $586,575$ instances in 6 domains. DomainNet can be considered a larger-scale dataset with a more difficult multi-classification task than the other 4 benchmarks.

## D  BASELINES AND IMPLEMENTATION DETAILS

**Baselines.**  To fairly compare our proposed framework with existing algorithms, we follow the settings of DomainBed (Gulrajani & Lopez-Paz, 2020) and DeepDG (Wang et al., 2022), using the best result between DomainBed, DeepDG, and the original literature. The comparisons include 12 baseline algorithms: ERM (Vapnik, 1999), IRM (Arjovsky et al., 2019), DRO (Sagawa et al., 2019), RSC (Huang et al., 2020), Mixup (Wang et al., 2020), MLDG (Li et al., 2018a), CORAL (Sun & Saenko, 2016), MMD (Li et al., 2018b), DANN (Ganin et al., 2016), C-DANN (Li et al., 2018d), DA-ERM (Dubey et al., 2021), and MIRO (Cha et al., 2022). Considering that domain labels can be leveraged as additional information for learning representations mitigating domain-specific features

Table 5: Hyper-parameters of DISPEL based on ERM.

| | PACS | Office-Home | VLCS | TerraInc | DomainNet |
|---|---|---|---|---|---|
| **DNN Architecture**: ResNet-18 | | | | | |
| Batch size | 128 | 128 | 128 | 128 | 128 |
| Learning rate | $1 \times 10^{-3}$ | $1 \times 10^{-3}$ | $3 \times 10^{-4}$ | $1 \times 10^{-4}$ | $1 \times 10^{-4}$ |
| $\tau$ | 0.1 | 0.1 | 0.1 | 0.1 | 0.1 |
| **DNN Architecture**: ResNet-50 | | | | | |
| Batch size | 64 | 64 | 64 | 64 | 64 |
| Learning rate | $5 \times 10^{-5}$ | $1 \times 10^{-3}$ | $1 \times 10^{-3}$ | $2 \times 10^{-4}$ | $1 \times 10^{-4}$ |
| $\tau$ | 0.1 | 0.1 | 0.1 | 0.1 | 0.1 |

projected to embedding space, we categorize the 12 baseline algorithms into two groups: **Group 1**: the algorithms requiring domain labels (Mixup, MLDG, CORAL, MMD, DANN, C-DANN, and DA-ERM); and **Group 2**: the algorithms without requiring domain labels (ERM, IRM, DRO, RSC, and MIRO).

**Implementation.** All the experimental results of the proposed DISPEL are implemented and performed based on the codebase of DeepDG (Wang et al., 2022). Unlike DomainBed (Gulrajani & Lopez-Paz, 2020), our implementation does not use any data augmentation during training. Regarding the setting of model selection, we use traditional *training-domain validation set* for our implementation, which does not require utilizing domain labels to split the desired validation set. For all the experimental results of DISPEL, we employ ERM algorithm to fine-tune the ResNet-18 and ResNet-50 as the fine-tuned model mentioned in Sec. 3.1. Concerning the use of the EMG, we utilize ResNet50 as the base model for EMG since the 5 domain generalization benchmarks we tested are image datasets.

**DNN Architectures.** The experimental results are all fine-tuned on the basis of ResNets. Since larger ResNets are known to have better generalization ability, we mainly conduct experiments with ResNet-50 models for all 5 benchmark datasets, and we also conduct the results of DISPEL based on ResNet-18 as a reference shown in Tab. 7. For both the two base network architectures, we both use the ResNet-18 and ResNet-50 pre-trained on ImageNet. As for the EMG component in DISPEL, we employ a ResNet50 pre-trained on ImageNet as the base model.

Table 6: Hyper-parameters of DISPEL for boosting other algorithms, where the DNN architecture is ResNet-50.

| | DRO | CORAL | DANN | Mixup |
|---|---|---|---|---|
| **Dataset**: PACS | | | | |
| Batch size | 64 | 64 | 64 | 64 |
| Learning rate | $1 \times 10^{-3}$ | $1 \times 10^{-3}$ | $5 \times 10^{-3}$ | $5 \times 10^{-4}$ |
| $\tau$ | 0.1 | 0.1 | 0.1 | 0.1 |
| **Dataset**: Office-Home | | | | |
| Batch size | 64 | 64 | 64 | 64 |
| Learning rate | $1 \times 10^{-3}$ | $1 \times 10^{-3}$ | $1 \times 10^{-3}$ | $1 \times 10^{-3}$ |
| $\tau$ | 0.1 | 0.1 | 0.1 | 0.1 |

## D.1 HYPER-PARAMETERS OF DISPEL

In the proposed DISPEL framework, the hyper-parameters are composed of batch size, learning rate, and $\tau$ in Eq. 1, where $\tau$ is the only hyper-parameter that is related to our algorithm. The hyper-parameters of DISPEL for each benchmark dataset are shown in Tab. 5.

## D.2 HYPER-PARAMETERS OF DISPEL FOR BOOSTING OTHER ALGORITHMS

As shown in Sec. 4.4, we leverage our DISPEL to further improve the prediction performance on unseen test domain for four existing domain generalization algorithms on PACS and Office Home,

Table 7: Each unseen test domain accuracy of DISPEL.

| | **Dataset**: PACS | | | | | |
|---|---|---|---|---|---|---|
| | Art Painting | Cartoon | Photo | Sketch | - | - |
| DISPEL (ResNet-18) | $83.6 \pm 0.3$ | $79.0 \pm 0.2$ | $97.0 \pm 0.0$ | $81.8 \pm 0.0$ | - | - |
| DISPEL (ResNet-50) | $87.1 \pm 0.1$ | $82.5 \pm 0.0$ | $98.0 \pm 0.1$ | $85.2 \pm 0.1$ | - | - |
| | **Dataset**: Office-Home | | | | | |
| | Art | Clipart | Product | Real | - | - |
| DISPEL (ResNet-18) | $61.4 \pm 0.0$ | $53.9 \pm 0.2$ | $76.0 \pm 0.1$ | $77.8 \pm 0.0$ | - | - |
| DISPEL (ResNet-50) | $71.3 \pm 0.5$ | $59.4 \pm 0.4$ | $80.3 \pm 0.3$ | $82.1 \pm 0.0$ | - | - |
| | **Dataset**: VLCS | | | | | |
| | Caltech101 | LabelMe | SUN09 | VOC2007 | - | - |
| DISPEL (ResNet-18) | $97.2 \pm 0.0$ | $62.6 \pm 0.1$ | $75.0 \pm 0.1$ | $76.9 \pm 0.1$ | - | - |
| DISPEL (ResNet-50) | $98.3 \pm 0.4$ | $65.3 \pm 0.1$ | $77.2 \pm 0.1$ | $76.3 \pm 0.1$ | - | - |
| | **Dataset**: Terra Incognita | | | | | |
| | Location 100 | Location 38 | Location 43 | Location 46 | - | - |
| DISPEL (ResNet-18) | $44.4 \pm 0.4$ | $49.6 \pm 0.7$ | $48.1 \pm 0.2$ | $37.3 \pm 0.1$ | - | - |
| DISPEL (ResNet-50) | $54.7 \pm 0.3$ | $48.1 \pm 0.0$ | $56.3 \pm 0.3$ | $42.3 \pm 0.2$ | - | - |
| | **Dataset**: DomainNet | | | | | |
| | Clipart | Infograph | Painting | Quickdraw | Real | Sketch |
| DISPEL (ResNet-18) | $44.6 \pm 0.0$ | $14.2 \pm 0.0$ | $39.7 \pm 0.0$ | $10.3 \pm 0.0$ | $45.6 \pm 0.0$ | $40.8 \pm 0.0$ |
| DISPEL (ResNet-50) | $63.4 \pm 0.0$ | $20.1 \pm 0.1$ | $48.2 \pm 0.0$ | $14.2 \pm 0.0$ | $63.4 \pm 0.0$ | $54.9 \pm 0.0$ |

Table 8: Each unseen test domain accuracy comparisons of Terra Incognita (ResNet50).

| | Location 100 | Location 38 | Location 43 | Location 46 |
|---|---|---|---|---|
| **Group 1**: algorithms requiring domain labels | | | | |
| Mixup (Wang et al., 2020) | $\mathbf{60.6} \pm 1.3$ | $41.1 \pm 1.8$ | $58.5 \pm 0.8$ | $35.2 \pm 1.1$ |
| MLDG (Li et al., 2018a) | $48.5 \pm 3.3$ | $42.8 \pm 0.4$ | $56.8 \pm 0.9$ | $36.3 \pm 0.5$ |
| CORAL (Sun & Saenko, 2016) | $48.6 \pm 0.9$ | $42.2 \pm 3.5$ | $55.9 \pm 0.6$ | $38.7 \pm 0.7$ |
| MMD (Li et al., 2018b) | $52.2 \pm 5.8$ | $47.0 \pm 0.6$ | $\mathbf{57.8} \pm 1.3$ | $40.3 \pm 0.5$ |
| DANN (Ganin et al., 2016) | $49.0 \pm 3.8$ | $46.3 \pm 1.7$ | $57.6 \pm 0.8$ | $40.6 \pm 1.7$ |
| C-DANN (Li et al., 2018d) | $49.5 \pm 3.8$ | $44.8 \pm 1.0$ | $57.3 \pm 1.1$ | $38.8 \pm 1.7$ |
| **Group 2**: algorithms without requiring domain labels | | | | |
| ERM (Vapnik, 1999) | $50.8 \pm 0.2$ | $42.5 \pm 0.2$ | $\mathbf{57.9} \pm 1.3$ | $37.6 \pm 1.3$ |
| IRM (Arjovsky et al., 2019) | $44.2 \pm 2.7$ | $41.3 \pm 0.6$ | $54.3 \pm 0.2$ | $36.0 \pm 1.7$ |
| DRO (Sagawa et al., 2019) | $31.8 \pm 0.3$ | $43.7 \pm 1.2$ | $58.0 \pm 0.7$ | $36.6 \pm 1.3$ |
| RSC (Huang et al., 2020) | $50.2 \pm 2.2$ | $39.2 \pm 1.4$ | $56.3 \pm 1.4$ | $40.8 \pm 0.6$ |
| DISPEL | $54.7 \pm 0.3$ | $\mathbf{48.1} \pm 0.0$ | $56.3 \pm 0.3$ | $\mathbf{42.3} \pm 0.2$ |

where all the DNN architectures are ResNet-50. The hyper-parameters of the DISPEL derivative models for the two datasets are shown in Tab. 6.

# E    EXPERIMENTAL RESULTS OF DISPEL

To closely investigate the fine-grained behavior of DISPEL in Sec. 4.3, we observe the prediction accuracy in each unseen test domain of all five domain generalization benchmark datasets. In Tab. 7, we show the experimental results of DISPEL on each unseen domain of five domain generalization benchmark datasets based on the two DNN architectures, ResNet-18 and ResNet-50. Based on the experimental results on each unseen domain, we conclude the **Observation 2: DISPEL possesses stable generalizing efficacy.** The results show that DISPEL maintains its stable efficacy in improving generalization ability over more different data distributions in more diverse classes of data. And these results reflect the purpose of the EMG module that considers each instance for fine-grained domain-specific feature masking.

# F    VISUALIZATION ANALYSIS VIA T-SNE

To illustrate how DISPEL improves generalization by blocking domain-specific features in the embedding space, we use t-SNE in the unseen test domains of all five benchmark datasets by comparing the embedding with and without DISPEL, as shown in Fig. 6 to Fig. 13. The key observation is that DISPEL aims to make each class more concentrated and separate them better. Taking PACS as an example, by drawing down more precise decision boundaries, the predictor can achieve better accuracy in the unseen *Art Painting* domain, in which DISPEL enhances the most accuracy among the 4 domains as shown in Tab. 2. Even in *Cartoon* domain where DISPEL only raises 0.7% accuracy, it shows the same intention to concentrate the embedding distribution for each class in Fig. 6-(b) and Fig. 7-(b). As for the unseen *Photo* domain, the base algorithm ERM has performed 96.7% accuracy, which means that Fig. 6-(c) reveals what a high-quality representation looks like. Compared to Fig. 7-(c), DISPEL follows the initial distribution and ameliorates the embedding to compress the distributions of each class.

Investigating the embedding of Terra Incognita, a more difficult multi-class classification task dataset, we observe the coherent behavior of DISPEL to its manner in PACS. As shown in Fig. 12-(a)(b)(d) and Fig. 13-(a)(b)(d), DISPEL has the same effect as on *Cartoon* and *Sketch* domain of PACS, which is to reduce the length of decision boundaries between different classes by concentrating distribution of each class. In addition, as shown in Tab. 8, *Location 43* is the only domain in which DISPEL cannot improve its classification accuracy. However, the reason is that we cannot achieve our reproduced ERM the same performance as provided in DomainBed (Gulrajani & Lopez-Paz, 2020), and the accuracy of our reproduced ERM in the *Location 43* domain is 55.1%. Therefore, DISPEL actually improves the accuracy in this unseen test domain by 1.3%. As we can see in Fig. 12-(c) and Fig. 13-(c), each class's instance embedding is concentrated after employing DISPEL as in other domains.

Based on the t-SNE visualization analysis, we conclude **Observation 3: DISPEL concentrate the distribution of each class embedding.** The t-SNE analysis demonstrates the superiority of DISPEL, which improves the domain generalization ability of the fine-tuned ERM by concentrating the distribution of embeddings in the same class.

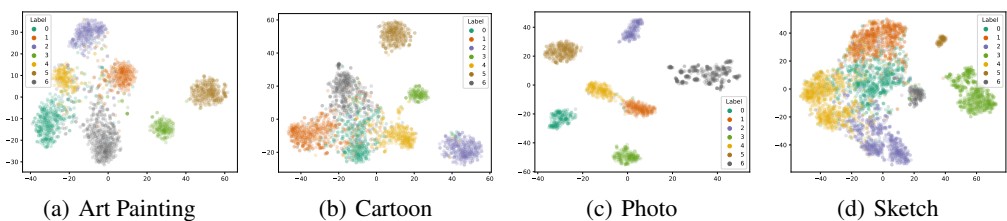

|    (a) Art Painting    |    (b) Cartoon    |    (c) Photo    |    (d) Sketch    |

Figure 6: t-SNE visualization of ERM embedding in four unseen test domains of PACS.

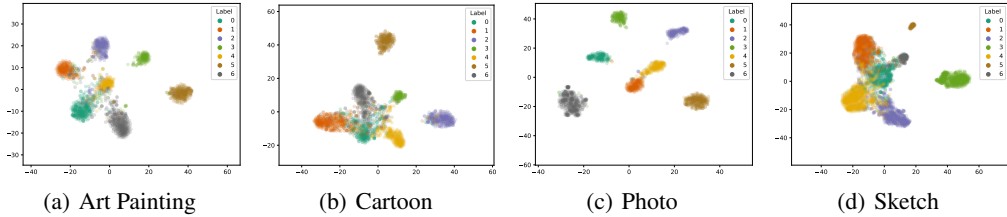

|    (a) Art Painting    |    (b) Cartoon    |    (c) Photo    |    (d) Sketch    |

Figure 7: t-SNE visualization of DISPEL embedding in four unseen test domains of PACS.

Furthermore, we conduct the quantitative results of the t-SNE average embedding variance over all classes in the PACS dataset. The results are shown in Table 9, where lower variances indicate more concentrated clusters of each class. The results demonstrate the efficacy and differences between a global mask and DISPEL.

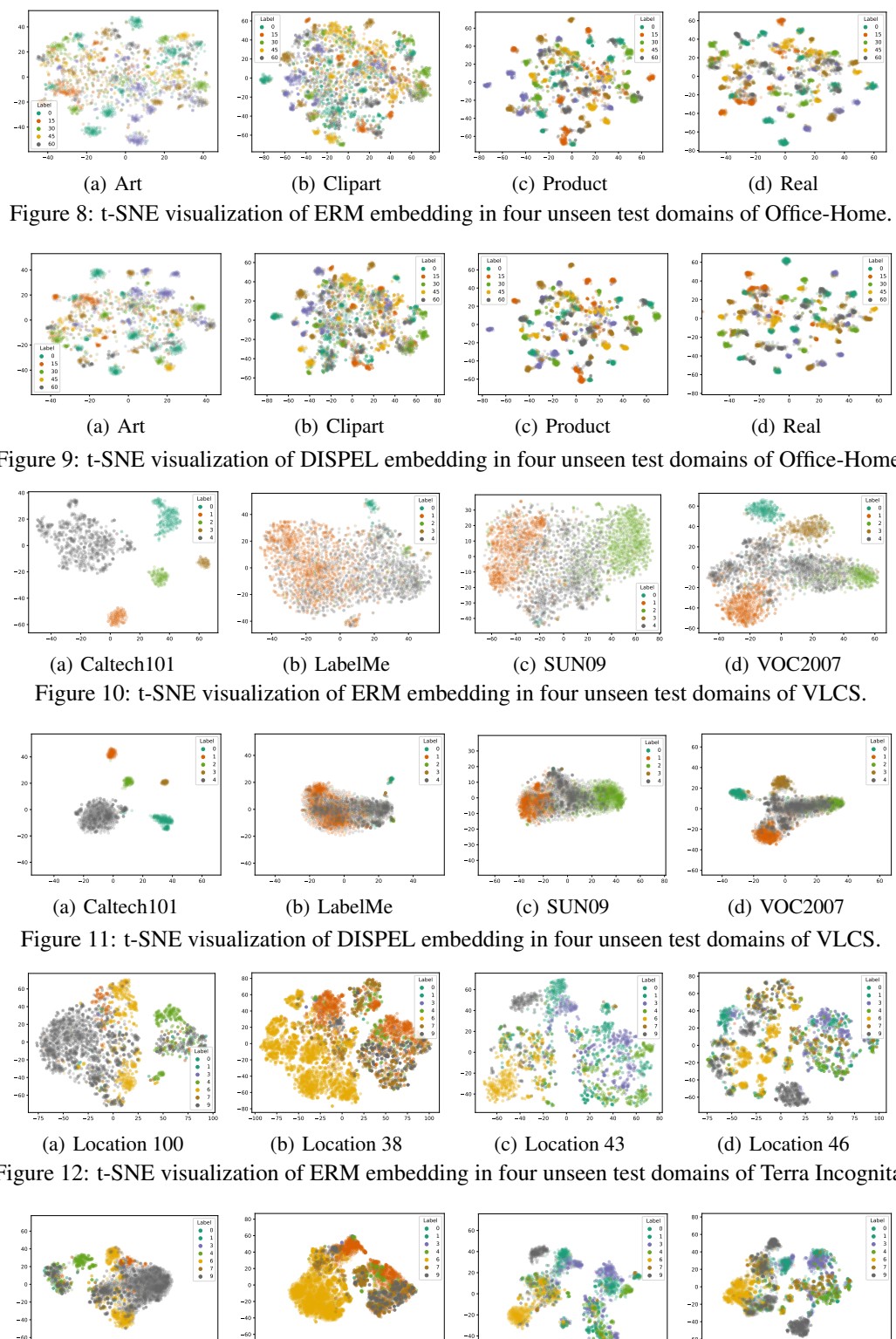

(a) Art   (b) Clipart   (c) Product   (d) Real

Figure 8: t-SNE visualization of ERM embedding in four unseen test domains of Office-Home.

(a) Art   (b) Clipart   (c) Product   (d) Real

Figure 9: t-SNE visualization of DISPEL embedding in four unseen test domains of Office-Home.

(a) Caltech101   (b) LabelMe   (c) SUN09   (d) VOC2007

Figure 10: t-SNE visualization of ERM embedding in four unseen test domains of VLCS.

(a) Caltech101   (b) LabelMe   (c) SUN09   (d) VOC2007

Figure 11: t-SNE visualization of DISPEL embedding in four unseen test domains of VLCS.

(a) Location 100   (b) Location 38   (c) Location 43   (d) Location 46

Figure 12: t-SNE visualization of ERM embedding in four unseen test domains of Terra Incognita.

(a) Location 100   (b) Location 38   (c) Location 43   (d) Location 46

Figure 13: t-SNE visualization of DISPEL embedding in four unseen test domains of Terra Incognita.

| Model | Art | Carton | Photo | Sketch | Avg. |
|---|---|---|---|---|---|
| ERM | 72.9 | 96.0 | 27.6 | 149.2 | 86.4 |
| Global Mask (Bottom 60%) | 55.8 | 59.4 | 21.6 | 122.1 | 64.7 |
| DISPEL | **37.6** | **46.8** | **20.8** | **51.2** | **39.1** |

Table 9: t-SNE average embedding variance (PACS).

