# OpenReview forum: "DISPEL: Domain Generalization via Domain-Specific Liberating"
_ICLR.cc/2024/Conference — ICLR 2024 Conference Withdrawn Submission_

### Official Review · Reviewer_Mwon · 2023-10-29

**Soundness:** 2 fair
**Presentation:** 3 good
**Contribution:** 2 fair
**Rating:** 3
**Confidence:** 5

**Summary:**

The paper discusses the challenge of domain generalization in machine learning, where a model needs to perform well on new, unseen domains after training on limited source domains. The authors propose a post-processing solution, called "Domain-specific Features Liberating (DISPEL)," that filters out indistinguishable domain-specific features using fine-grained masking in the embedding space. The authors demonstrate the performance of DISPEL on five benchmarks and provide a theoretical framework for the generalization performance of the proposed method.

**Strengths:**

The authors propose a simple method in order to improve the performance of the pre-trained model on unseen test domains. The method demonstrates stable improvement by few percent across almost all benchmarks. The method doesn't require labels from source domains.

**Weaknesses:**

The main weakness of the paper is the theoretical justification of the method. Unfortunately, Theorem 1 has nothing to do with the generalization error on unseen domains and the original definition of the generalization error in the first place. Proper definitions could be for example found in the works of Ben-David (e.g. "A theory of learning from different domains").

**Questions:**

- The logic of splitting methods in group 1 and group 2 is not very clear to me. These methods could be run using labels only in source domains as discussed in DomainBed paper. Could the authors please clarify that?
- It is quite interesting that pretrained masked generator doesn't suffer from generalization issues when applied to new unseen domains. Do the authors have an intuition why this could be the case?
- It is an interesting property, that the method doesn't require labels from source domains. However, it is a bit difficult for me to imagine a situation when you would still have access to the full source inputs, but not to labels (given that you did have access to labels in order to pre-train the network in the first place). The authors stress it several times throughout the paper that this is an important property of the proposed approach, so it would be good if they could clarify this point.
- What is the numerical difference of the proposed method compared to global masking? Could the authors please provide the table with the ablation study?
- It looks like the authors inexplicitly assume that the embedding dimensions are disentangled with respect to domain-specific and domain-sharing features. For me it is not obvious why this would be a general case. Could the authors comment on that?

---

### Official Review · Reviewer_u3X2 · 2023-10-31

**Soundness:** 2 fair
**Presentation:** 3 good
**Contribution:** 2 fair
**Rating:** 5
**Confidence:** 3

**Summary:**

The authors propose a model for domain generalization without domain labels, DISPEL, which is a post-processing feature-masking approach that can filter out domain-specific features in the embedding space.
The authors derive a generalization error bound (but is not like a conventional generalization bound in the sense of statistical learning theory, which often includes sample-size dependence) to guarantee the generalization performance.
DISPEL achieves SOTA performance in some cases.

**Strengths:**

- The proposed method is simple but effective. The proposed method achieves state-of-the-art performance in some cases, even without leveraging domain labels and any data augmentation method.

- The experiments are nicely designed. Error bars are given. The experimental results are mostly significant.

- Source code is available, which is critically important in domain generalization, where reproducibility is a crucial problem.

- The paper is easy to follow.

**Weaknesses:**

- (Critical) The underlying mechanism, i.e., why DISPEL works well in domain generalization, is not completely clear (see Questions). I would like to know more about why DISPEL works well in unseen target domains.

- The main idea of decomposing features into domain-invariant and domain-specific features has been explored and is not novel.

**Questions:**

- [Question (major)] The idea of the proposed masking looks similar to other methods that attempt to obtain domain-invariant features (in fact, some call domain-sharing feature as domain-invariant feature in the literature). The question is: What is the novelty of the proposed method compared to the previous methods that aim to obtain domain-invariant features? Please show an evidence that DISPEL is not a "yet another model" in domain generalization.

- [Question (major)] Why is the temperature-dependent Gumbel softmax (and the associated sampling of random variables) used? Why are they necessary? There are many other choices, e.g., a simple softmax, etc.

- [Question (major)] Is there any ablation study about Section 3.2?

- [Question (major)] EMG is designed to remove domain-specific features, but the masks thus obtained does not know the target domain data anyway, which means they may overfit to source domain data. Therefore, I guess the performance improvement in experiment comes from the randomness introduced in the Gumbel softmax. So I have a question: Is the mask generation process in the inference phase deterministic? If no, please show the variance due to the randomness. If yes, how did you fix the randomness?

- [Question (major)] In Section 3.4:
> DISPEL framework minimizes the value of second terms in Eq. 1 by making z^T-sp_k approaches z^T-sp_k with a generated mask.

Could you elaborate on this statement? What do you mean by "second terms in Eq. 1"? How does DISPED framework minimizes what?

---

### Official Review · Reviewer_StPi · 2023-10-31

**Soundness:** 3 good
**Presentation:** 2 fair
**Contribution:** 2 fair
**Rating:** 3
**Confidence:** 4

**Summary:**

This work tries to develop a domain generalization method that needs no data-augmentation technique or domain label information. It is a post-processing fine-grained masking approach that learns to filter out domain-specific features for each sample separately in the embedding space. The authors provide theoretical and experimental analysis to verify their method.

**Strengths:**

The method proposes a kind of sample-level domain-specific feature identification which is supposed to be more flexible than existing global-level method considering the complex domain information in real application.

**Weaknesses:**

There are many publications also try to solve DG from the view of separating the representation into domain specific and domain invariant parts, which is not mentioned in this work. The analysis about the mask learning is not sufficient to support their idea.

**Questions:**

1.	The authors claim that “data manipulation methods require highly engineered efforts and can introduce too much prediction-irrelevant knowledge label noise”, please provide more explanation about the " prediction-irrelevant knowledge label noise ", and how it influence the generalization performance.

2.	Some important closely related work are not mentioned in this paper, for example, to divide the representation into domain-specific and domain invariant, some work use the disentangling technique, and some work use causal inference. And what about the performance compared to these methods?
3.	equation (1), what are i and j for? It looks like they are not related to the right side of the equation? What’s the effect of different /tao on the generated masks? And how to set \tao to avoid the trivial outcome?

4.	How does equation(3) make sure that the masked part of the embedding is domain-specific? Since when the masked part of the embedding learns something redundancy or constant, equation(3) can still be minimized

5.	what does the final masks look like, is there any obvious patten in the generated masks which is highly related to domain information? If so, there will be a pattern highly domain related in the embeddings after the mask operation, which is contradictory to the goal of domain invariant.

6.	Does the author use any traditional data-augmentation techniques, (e.g. random crop, random flip,…) in the ERM pre finetuning period?

7.	there is no analysis on the effect of the sparseness of the mask. How to choose a proper parameter to control it given a specific dataset?
8.	what about the performance when compared to some SOTA method, e.g. SWAD?

---

### Official Review · Reviewer_zEDH · 2023-11-06

**Soundness:** 3 good
**Presentation:** 3 good
**Contribution:** 3 good
**Rating:** 6
**Confidence:** 4

**Summary:**

This paper proposes a feature learning based method for domain generalization under assumptions that there exists two types of features: i) domain-shared and ii) domain-specific features, and a good out-of-distribution performance can be achieved by filtering out the domain-specific features. One way to do so is to apply a global masking on the top of the features to retain only the domain-shared features. It is empirically shown that the global masking slightly improves the domain generalization performance, but the result is still sub-optimal.

To better optimize the impact from the masking, a method for domain generalization namely DomaIn-SPEcific Liberating (DISPEL) is proposed, which introduces a learnable mask called Embedding Mask Generator (EMG) that automatically constructs a distinct filter for each input data. DISPEL finds an optimal EMG that minimizes the cross entropy between the normal and feature-masked prediction while keeping the foundational model frozen.

The performance evaluation on various benchmarks empirically demonstrates the effectiveness of DISPEL, producing state-of-the-art performance. The theoretical analysis provides a guarantee that DISPEL improves the domain generalization performance.

**Strengths:**

$\textbf{Performance and practicality}$: DISPEL consistently produces superior domain generalization performance over existing methods on 5 benchmarks, with a relatively simple implementation, i.e., applying trainable masking in the embedding space. Furthermore, it does not require domain label information and can be used additively with any deep learning based finetuning method as demonstrated in Table 4.

$\textbf{Novelty}$: To my knowledge, this is the first attempt in employing dynamic, instance-specific feature masking in the context of domain generalization.

**Weaknesses:**

$\textbf{Performance Stability}$: As stated in Sec 4.3.1, DISPEL possesses stable generalizing efficacy (Observation 2). If the “stability” here means that there is no significant performance degradation from across a wide range of benchmarks, I’m not sure if the empirical evaluation is convincing enough to support the claim. I would argue that, for examples, Mixup and MIRO are “stable” enough.

Furthermore, the early premise about the needs of DISPEL as a more stable solution over the global masking (since it’s stated that its effectiveness varies for different compositions of training domains – hence it’s less stable) is not backed with empirical evidence.

I would like to hear more from the authors regarding this concern.

$\textbf{Comparison with recent work}$: A few recent work by, such as [Ding et al. NeurIPS 2022] and [Bui et al. NeurIPS 2021] that also improve domain generalization by devaluing domain-specific features are not discussed. I encourage that the empirical comparison with those methods using the same benchmarks should be included.

[Ding et al. NeurIPS 2022] Domain Generalization by Learning and Removing Domain-specific Features

[Bui et al. NeurIPS 2021] Exploiting Domain-Specific Features to Enhance Domain Generalization

Some minor issues from my side are as follows:
-	Putting the indices explicitly at the corresponding terms in equation (3) would be helpful.
-	Subsection 4.3.1: DoainNet  DomainNet
-	Theorem 1: I think the notations (i.e., superscript parts) for denoting domain-specific and domain-sharing features are interchanged?

**Questions:**

Is it necessary to use the base model as large as ResNet50 for the mask generator? Would much smaller models work?

How are the convergence behaviour and time complexity of the EGM training?

Why are DA-ERM and MIRO excluded from the evaluation comparison in Table 2?